# Origin of large plasticity and multiscale effects in iron-based metallic glasses

Baran Sarac [1], Yurii P. Ivanov [1,2], Andrey Chuvilin [3,4], Thomas Schöberl[1], Mihai Stoica[5,6], Zaoli Zhang[1] & Jürgen Eckert[1,7]

The large plasticity observed in newly developed monolithic bulk metallic glasses under quasi-static compression raises a question about the contribution of atomic scale effects. Here, nanocrystals on the order of 1–1.5 nm in size are observed within an Fe-based bulk metallic glass using aberration-corrected high-resolution transmission electron microscopy (HRTEM). The accumulation of nanocrystals is linked to the presence of hard and soft zones, which is connected to the micro-scale hardness and elastic modulus confirmed by nanoindentation. Furthermore, we performed systematic simulations of HRTEM images at varying sample thicknesses, and established a theoretical model for the estimation of the shear transformation zone size. The findings suggest that the main mechanism behind the formation of softer regions are the homogenously dispersed nanocrystals, which are responsible for the start and stop mechanism of shear transformation zones and hence, play a key role in the enhancement of mechanical properties.

[1] Erich Schmid Institute of Materials Science, Austrian Academy of Sciences, 8700 Leoben, Austria. [2] School of Natural Sciences, Far Eastern Federal University, Vladivostok 690950, Russia. [3] CIC nanoGUNE Consolider, 20018 San Sebastian, Spain. [4] IKERBASQUE, Basque Foundation for Science, 48013 Bilbao, Spain. [5] Laboratory of Metal Physics and Technology, Department of Materials, ETH Zurich, 8093 Zurich, Switzerland. [6] Politehnica University of Timisoara, 300006 Timisoara, Romania. [7] Department of Materials Physics, Montanuniversität Leoben, 8700 Leoben, Austria. Correspondence and requests for materials should be addressed to B.S. (email: baran.sarac@oeaw.ac.at)

Despite ultra-high strength and elastic energy absorption capacity, the negligible plasticity in most metallic glasses has been an important barrier obstructing property optimization and their widespread use[1,2]. Yet, some novel bulk metallic glass (BMG) forming systems developed in the last two decades have overcome this barrier by often showing extreme compressive plasticity (>15%) along with considerably high yield strength[3–6]. The extent of deformation in these specific alloys has been linked to the homogenous and concurrent nucleation and evolution of a high-density of shear bands (SBs) throughout the volume and their intersection with each other. Nevertheless, compared to their counterparts with detectable second phase crystallinity and/or devitrification in their pre-deformed state[7–11], as well as generated upon deformation (in which plastic shear is absorbed or deflected by the homogeneously dispersed second phases or voids)[12–17], the underlying intrinsic mechanism for achieving large plasticity for these monolithic BMGs is still unclear. Liu et al. linked the generation of high plasticity with hardness fluctuations on the micron-scale originating from structural heterogeneities of the same composition as the glassy phase within the BMG matrix[3,18]. However, the extreme plasticity observed in these newly developed BMGs raises questions about the contribution of atomic-scale effects, which necessitates further investigations on the atomic-scale to address the origin of these soft zones dispersed homogeneously within the hard phase. Despite the efforts to elucidate the atomic configuration and bond strength between atoms, an atomic resolution imaging is required to investigate the local heterogeneities in metallic glasses.

The new generation microscopes with spherical aberration correction (Cs-corrector) and monochromator have marked a new era in the field of electron microscopy with sub-eV energy resolution and sub-Angstrom spatial resolution[19,20]. Using modern aberration-corrected high-resolution transmission electron microscopy (HRTEM) imaging, the medium-range-order/nanocrystalline clusters embedded in the amorphous phases were captured in real space[21–23].

Among amorphous alloys, Fe-based BMGs are of interest due to their soft magnetic properties with low loss, large permeability, and high-saturation magnetization, which makes them attractive for electro-magnetic conversion devices for power electronics (e.g. relays, automatic switches, circuit breakers, transformers) at high frequencies[24–27]. The main reason for these appealing properties arises from the amorphous nature with the absence of crystalline anisotropy[28]. Besides, it has been reported that a newly developed 1 mm diameter $Fe_{50}Ni_{30}P_{13}C_7$ BMG possesses extensive plasticity (up to 22% plasticity) together with a fracture toughness of $K_C \approx 50$ MPa m$^{-1/2}$ with a yield strength of $\sigma_y = 2250$ MPa[5], where the yield strength and plastic strain with 2 mm diameter samples are 2800 MPa (measured from the deviation point of linearity) and 3%[29]. This implies that the cooling rate has a strong impact on the mechanical behavior, which serves as an additional motivation of the present investigation. The unexpectedly favorable mechanical properties of Fe-based BMGs containing metalloid were linked to the transfer of the valence electrons from the $p$-shell of the metalloid elements to the $d$-shell of the transition metal, rendering covalent $p-d$ hybrid bonding that can resist shear dramatically[30,31].

The main goal of this study is to resolve the origin of large plasticity, which is the nano-scale heterogeneities, from an atomic- and micro-scale point of view using a combination of an advanced TEM and nanoindentation techniques. High-resolution HRTEM complemented by TEM image simulations at varying sample thicknesses are conducted to reveal structural modifications at the atomic length-scale. We show that the extent of plasticity in monolithic glasses is linked with the existence of homogenously dispersed nanocrystals formed during fast quenching rates which are responsible for the initiation and impediment of the plastic deformation. Aberration-corrected HRTEM technique used in this study has a point-to-point resolution less than 0.1 nm, which allows us to acquire easily an image of nanocrystal in sizes of 1–1.5 nm in the amorphous matrix. Therefore, the findings of this study give deeper insight to Yang et al.'s paper on the extensive plasticity observed for the considered Fe-based BMG[7]. In addition, this paper aims for an understanding of the role of the extremely small crystals and their nano- and micro-scale dispersion on the overall large deformation, which brings a fresh perspective to the deformation mechanism in other BMG classes exhibiting extensive plasticity.

## Results

**Mechanical characterization.** Figure 1a depicts the macroscopic deformation behavior of the $Fe_{50}Ni_{30}P_{13}C_7$ BMG. Average yield strength measured from the point of deviation from linear elastic behavior is $\sigma_y = 2265 \pm 100$ MPa. Large fluctuations in terms of the fracture strain (>50.0%, 45.6%, and 26.3% for the blue, red, and black curves, respectively) is observed which is linked with the minor differences between the samples (e.g. aspect ratio, parallelism between the top and bottom side, casting surface

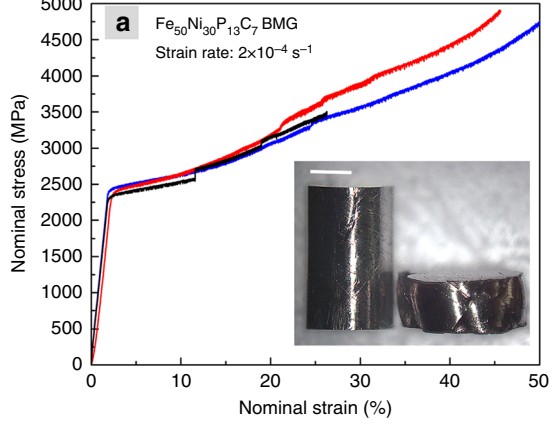
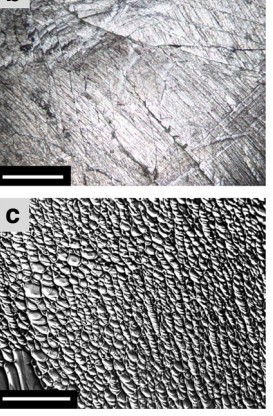

**Fig. 1** Mechanical characterization of $Fe_{50}Ni_{30}P_{13}C_7$ BMG. **a** Nominal stress vs. strain curves of the deformed BMG at a quasi-static strain rate of $2 \times 10^{-4}$/s. Inset shows the samples before and after deformation (up to 50% strain, scale bar 500 μm). Small rises in the nominal stress (particularly in the black curve) are possibly due to sudden local fractures which temporarily change the deformation plane. **b** Deformation surface of the compressed BMG (sample belonging to the blue curve) showing multiple shear bands (scale bar 100 μm). **c** The fracture surface corroborates the large plastic deformation of this BMG (scale bar 100 μm)

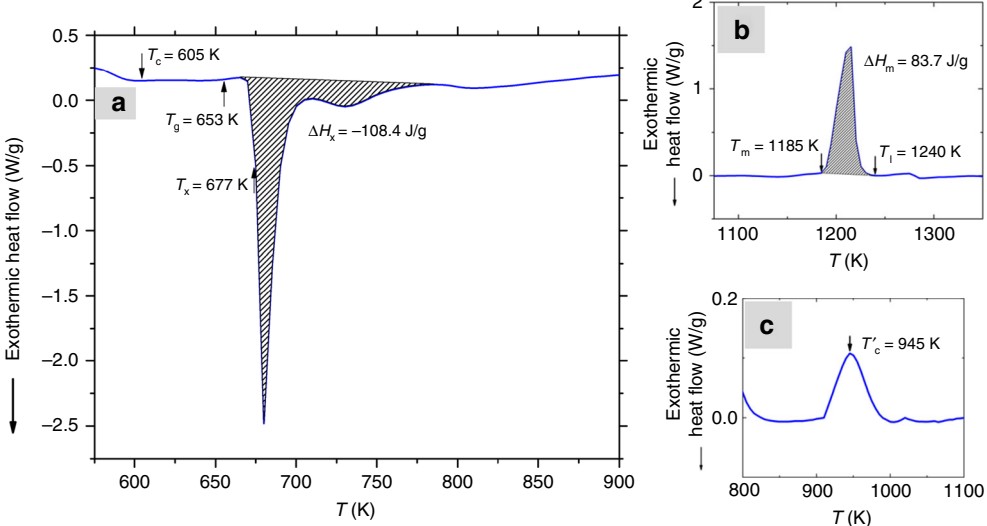

**Fig. 2** Thermal properties of $Fe_{50}Ni_{30}P_{13}C_7$ BMG. The glass transition $T_g$, crystallization temperature $T_x$, Curie temperature $T_c$, and heat of crystallization $DH_x$ are provided in (**a**). The melting temperature $T_m$, liquidus temperature $T_l$, and heat of melting $\Delta H_x$ is provided in (**b**). In (**c**), Curie temperature $T'_c$ of the Curie temperature of the magnetic crystalline phases formed during the quenching and/or heating step is shown

defects, distance of the sample from the center of the compression fixtures)[32,33]. Severe deformation can be observed specifically for the samples indicated by blue and red curve. The height of the sample decreases from 2 to 0.8 mm, and the compressed BMG shows extensive shear and cleavage (inset to Fig. 1). In the vicinity of these major shear zones, finer SBs intersect with each other (shown in Fig. 1b). The SEM investigation of the fracture plane presents the dimpled fracture surface confirming the existence of severe deformation (Fig. 1c).

**Thermal characterization**. DSC curve for the 1-mm water-cooled and fluxed copper mold casting of the $Fe_{50}Ni_{30}P_{13}C_7$ BMG is given in Fig. 2. The extent of the supercooled liquid region (Fig. 2a), $\Delta T = T_x - T_g$, is 24 K confirming the previous findings[5,34]. The Curie temperature of the amorphous phase $T_c$ is measured as 605 K, while the enthalpy of crystallization and melting are measured as $\Delta H_x = -108.4$ J/g (Fig. 2a) and $\Delta H_m = 83.7$ J/g (Fig. 2b), respectively. No additional shoulder upon melting is observed indicating a possible homogenous single phase melt. A clear Curie temperature of the magnetic crystalline phases $T'_c$ is also observed at 945 K (Fig. 2c).

**Investigation of microscale heterogeneities**. Figure 3a shows low-magnification STEM micrographs of a $Fe_{50}Ni_{30}P_{13}C_7$ BMG sample prepared from the lateral cross-section of a 1 mm diameter cast rod. As expected from our casting conditions as well as the literature findings[5,29], the structure is amorphous with no signs of crystallinity on the meso-scale. The difference in thinning rates (produced under low argon ion beam energy and liquid nitrogen temperature) generates bright and dark zones imaged from different regions. Similar to ref. [3], the regions do not show detectable phase separation as confirmed by energy dispersive spectroscopy. During ion-milling, liquid nitrogen cooling was used to eliminate any kind of artifacts caused by this process (e.g. nano-segregation, crystallization)[35]. Nanoindentation tests were conducted on the mechanically polished disc specimens at room temperature with a maximum load of 10 mN using a standard Berkovich indenter tip. Due to the thermal drift of the nanoindenter, the positions of the indents have deviations on the order of ±0.05 mm. Therefore, a minimum distance of 0.1 mm was kept between the indents. Further details of the indentation procedure

and evaluation are given in the last chapter, Methods. The nanoindentation results from 20 different positions (marked by white X) are shown as contour plots of hardness and reduced modulus in Fig. 3b. The distances are in mm, and the center of the disc is denoted by 0. Corresponding load−displacement curves are shown in Fig. 3c. Significant alterations between the indentation depths at constant maximum load of 10 mN (from 149.6 nm to 186.7 nm) as well as a remarkable drop in the elastic modulus ($\Delta E(\%) = (222.1 − 168.1)/222.1 \times 100 = 24.3\%$) were recorded (Fig. 3d). The hardness fluctuates between 6.15 and 8.57 GPa (inset in Fig. 3d). The average hardness is $6.84 \pm 0.56$ GPa, the corresponding Tabor factor results in a value close to 3. The lateral extension of the plastic zone varies, depending on the model in use, between 1.4 and 4 μm, the vertical extension being about 1.7 μm[36,37]. Our results confirm the previous findings that the difference between the bright and dark zones is linked to preferential thinning during sample preparation[3,18,38]. To bring insight to the origin of this effect we performed HRTEM studies.

**HRTEM investigation of nanocrystals**. Figure 4a, f shows typical HRTEM images of the specimen with an average thickness of 8 nm (measured by analyzing the zero-loss peak of the electron energy-loss spectroscopy (EELS)[39]). While the specimen is mostly amorphous, there are areas with local atomic ordering. Figure 4b, d, g, i shows magnified regions with and without local atomic arrangement. The lattice fringes are produced by interference of the transmitted and diffracted beams and observed on the crystals if the lattice spacing is larger than the resolution of the microscope. The local ordered atomic arrangement areas correspond to nanocrystals (NCs) of 1–1.5 nm size originated during casting and homogenously spread throughout the specimen. FFT patterns of the corresponding regions (Fig. 4c, h) show bright spots on the broad halo indicating the presence of these NCs. The average lattice fringes are around $2.05 \pm 0.05$ Å, which may correspond to an FCC metallic crystalline phase (FCC-FeNi with bulk (111) lattice spacing of 2.08 Å)[40]. The matrix phase exhibits a maze-like amorphous pattern (Fig. 4d, i) with broad diffuse diffraction halo with no point maxima in the FFT diffraction pattern (Fig. 4e, j). To study quantitatively the atomic-scale structural ordering, the HRTEM images were analyzed using custom DigitalMicrograph scripts[41] based on evaluation of the

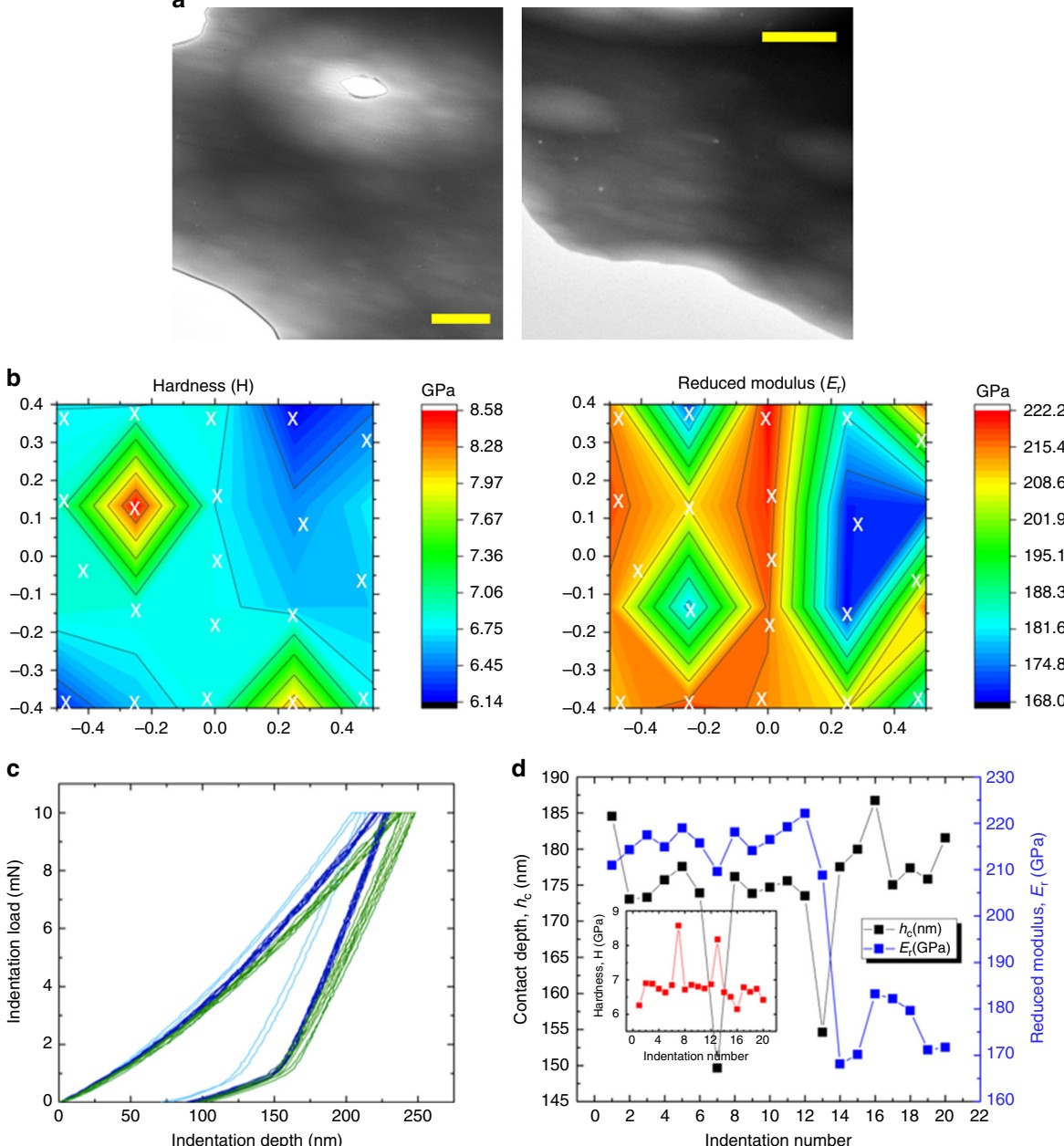

**Fig. 3** Investigation of microscale heterogeneities. **a** Low-magnification STEM images taken from the cross-section of the Fe$_{50}$Ni$_{30}$P$_{13}$C$_7$ BMG. In order to eliminate any oxide layer and microcrystallinity generally forming on the surface during casting, our investigations were performed at the center region of the specimen (scale bars 2 μm). **b** Hardness & Reduced Modulus contour plots of 1 mm-sized Fe-BMG disc (the distances are in mm). **c** Indentation load vs. indentation depth of the corresponding measurements. **d** The variations in the contact depth, reduced modulus, and hardness (inset) obtained from the nanoindentation tests

local autocorrelation function. Autocorrelation analysis gives a quantitative estimate of the local atomic arrangement and creates a 2D map of this estimate—higher value reflects the better periodic ordering of the atoms. This 2D map is used as a mask overlaid on the image highlighting the locations of the nanocrystals. Nanocrystalline clusters (volume fraction of about 20 vol. %) are marked by red color on the HRTEM image presented in Fig. 5a, where the mask used for the calculation is shown in Fig. 5b. Such concentration of crystal-like structures observed in Fig. 5a (red color) is probably responsible for the local softening found in Fig. 3d. The zones that are etched faster are relatively softer (with lower $H$ and/or $E_r$) compared to the darker regions probably due to the abundance of metal–metal pairs. The

energy-loss spectrum for the Fe$_{50}$Ni$_{30}$P$_{13}$C$_7$ BMG from the same thin regions is depicted in Fig. 5c. $I_t$ (total number of number of electrons in the EELS) is the entire area of the peak above the dashed blue line. $I_0$ (number of number of electrons having the zero–loss peak) is the red-hatched area indicating the sharp peak. The characteristic mean free path for the inelastic scattering of the material, $\lambda \approx 100$ nm, is calculated from the effective atomic number of the adopted Fe-based BMG, which is in accordance with the results found for the Fe and its alloys[39]. Thus, using the equation:

$$\frac{t}{\lambda} = \ln\left(\frac{I_t}{I_0}\right) \tag{1}$$

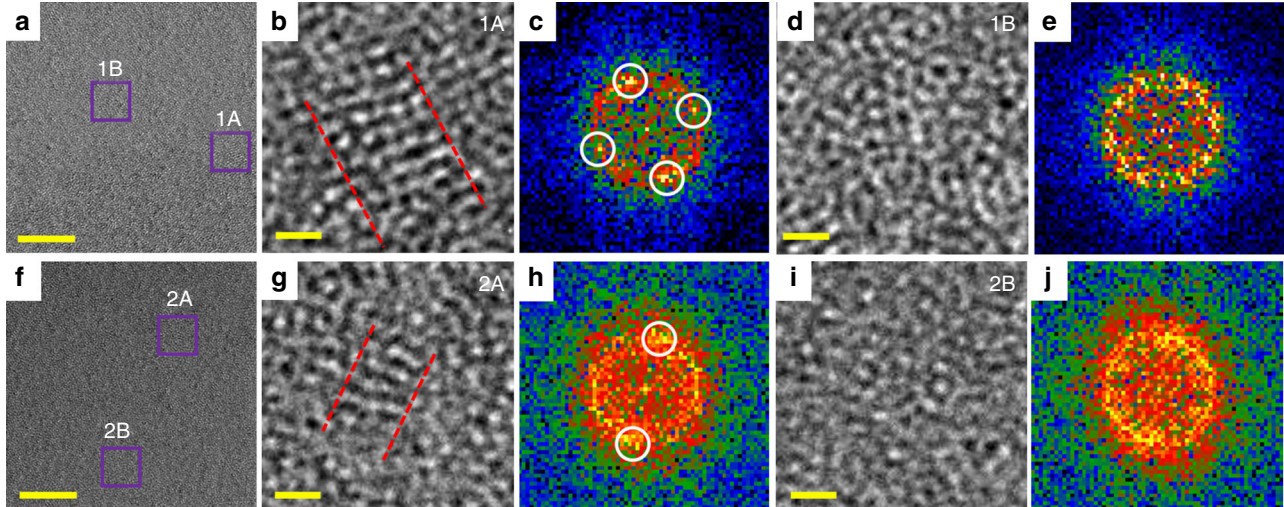

**Fig. 4** HRTEM investigation of nanoscale heterogeneities. **a**, **f** 1A and 2A regions show nanocrystals in the amorphous matrix (1B and 2B). **b**, **d**, **g**, **i** HRTEM images and **c**, **e**, **h**, **j** corresponding FFT patterns taken from two different regions of the specimen with 8 nm thickness. . The spots encircled by white circles in the FFT patterns in (**c**) and (**h**) corroborate the presence of nanocrystals in the specimen, whereas no speckles are observed in the FFT patterns in (**e**) and (**j**)  (scale bars, **a** and **f**—5 nm, **b**, **d**, **g**, and **i**—0.5 nm)

the thickness of the sample can be estimated as 8 nm. Figure 5d shows the filtered HRTEM image (for better visualization of the crystal-like areas) after autocorrelation analysis. In inserts FFT patterns for the areas of different sizes are shown. FFTs from the small regions (comparable to the NC size) 1–2 show pronounced lattice fringes. The larger areas 3–4 depict pronounced amorphous ring from the matrix that is strongly affected on the visualization of the fringes from nanocrystals. Similar effect is shown in the literature by Fluctuation Electron Microscopy (nanodiffractions); only at the beam sizes comparable to the area of interest the effect of the medium range order is clearly visible[42–44]. The collection of the FFTs only from the areas with nanocrystals shows more pronounced lattice fringes on the resulted FFT pattern. On the other hand, SAED analysis area is typically at least 100 nm, which would cover multiple fields of view of the image on Fig. 5a, so the diffraction spots from the nanoclusters are rotationally averaged and spots do not show up. Figure 5e confirms the fully glassy state across the specimen with no crystallization peak observed. In addition, we probed various regions in the sample where different distribution of the nanocrystal clusters after autocorrelation analysis of FFT patterns are observed. The fraction of the nanocrystal clusters taken from an average thickness of 8 nm from the bright zones in LM STEM (Fig. 5f) is approximately 20 vol.%. Relatively darker zones in LM STEM having lamellae thickness greater than 8 nm but below 20 nm (Fig. 5g) have nanocrystal cluster concentration as low as 10 vol.%. The clusters are more homogenously dispersed in the thinner regions than the thicker ones.

For better understanding the HRTEM images presented in Fig. 4 we also performed systematic simulations of the HRTEM images of the 1-nm-sized NCs embedded inside the amorphous matrix. By image simulation we can determine the thickness range at which the lattice fringes from such small crystals inside amorphous material are visible. The structure model of the $Fe_{50}Ni_{30}P_{13}C_7$ BMG was obtained from molecular dynamics simulations (MD) and is shown in Fig. 6a. A dense random packing model with up to 30,000 atoms was constructed from melting an iron phosphide nickel mineral (Schreibersite with I-4 space group) with a composition of $Fe_{2.1}Ni_{0.9}P$ at 1800 °C. Then, FeNi fcc crystals (lattice constant of ~0.36 nm)[45] with a diameter of about 1 nm were embedded in the middle part of the model.

TEM simulations were conducted for samples with 4 nm to 23 nm thickness. Figure 6b–d shows the selected HRTEM image simulations performed from the final structure model for the regions of 4 × 4 × 8 nm, 4 × 4 × 15 nm, 4 × 4 × 23 nm and their FFT patterns (Fig. 6e−g), respectively. The lattice fringes of the NC are clearly visible in the wide range of defocus values used for simulation (±20 nm) only up to 10 nm thickness of the amorphous phase. For thicker models, a strong decay of the contrast from fringes was observed. At 23 nm thickness the lattice fringes from NCs are no longer visible. The findings agree well with our HRTEM investigations and highlight the difficulties for the estimation of the nanocrystal volume fraction from thicker regions.

## Discussion

The idea of generating local structural heterogeneity in the matrix was achieved by partially replacing iron atoms with nickel. The enthalpy of mixing of Fe−Ni, $\Delta H_{Fe-Ni}^{mix}$, is very small (−2 kJ/mol) as compared to $\Delta H_{Ni-C}^{mix} = -39.5\,kJ/mol$ and $\Delta H_{Fe-C}^{mix} = -50\,kJ/mol$[46], rendering a low atomic bond force between Fe and Ni atoms. Besides, the enthalpy of mixing between $Ni$ and metalloids is $\Delta H_{Ni-P}^{mix} = -34.5\,kJ/mol$ and $\Delta H_{Ni-C}^{mix} = -39\,kJ/mol$[46], respectively. Hence, metal (Fe) vs. metalloid (C and P) atom pairs generate the amorphous matrix, whereas Ni atoms are not incorporated in these medium-range order nanoclusters[5]. The metal−metalloid bonds are known to exhibit highly directional covalent bonds that can resist deformation whereas metal−metal bonds have lower directionality thus accommodating shear[47]. This can certainly contribute to the explanation of why remarkable plasticity can be observed for Fe-based BMGs with high metal content[25,30,48]. Recent observations for the same composition have postulated that the thermal parameters such as the width of the super-cooled liquid region $\Delta T_x (= T_x - T_g)$, the reduced glass transition temperature $T_{rg} (= (T_g)/(T_l))$ and the $\gamma$ parameter $(= (T_x)/(T_g + T_l))$ defining the glass-forming ability (GFA) of the BMG decrease for more than 5 at.% Ni addition[34]. Another important point is that Fe and Ni are both late transition metals with metallic radii of 126 and 124 pm, respectively[49]. Hence, the difference between the atomic radii is less than 12%, and thus

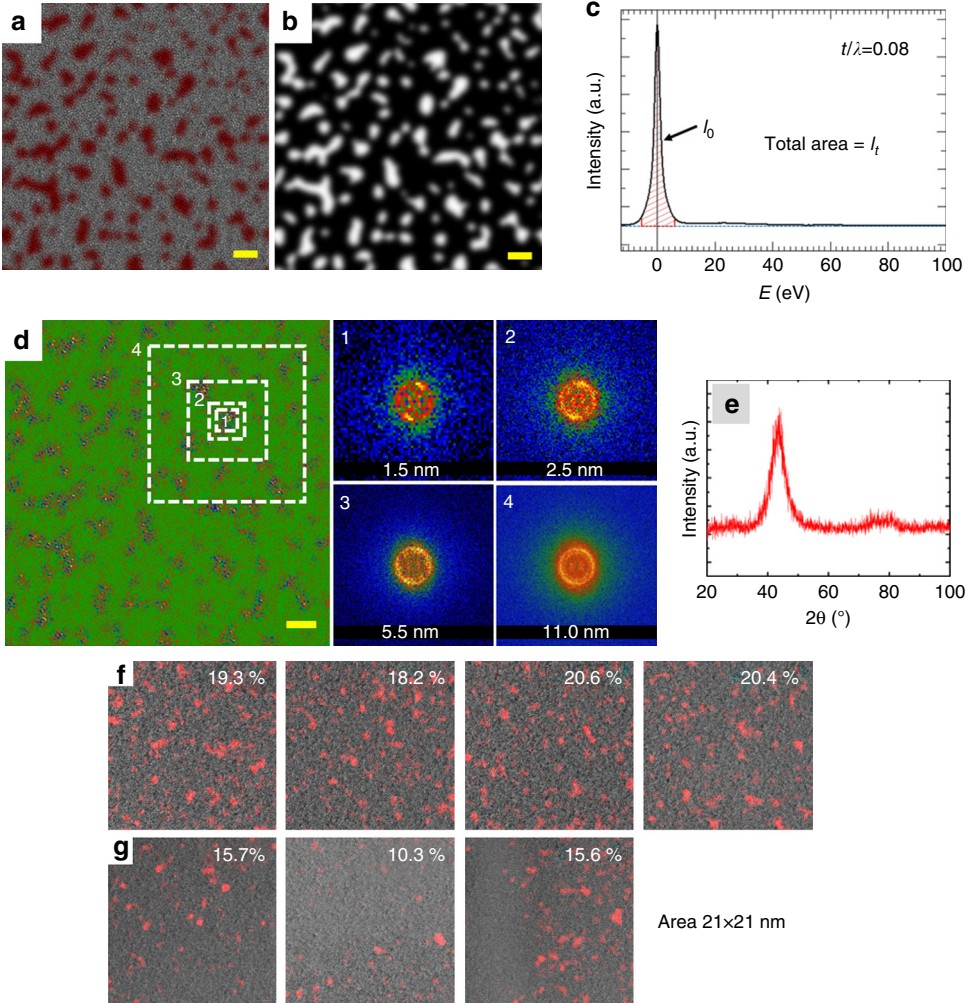

**Fig. 5** Crystal-like cluster determination. **a** Mapping of the local atomic arrangement retrieved by autocorrelation analysis of the HRTEM image in Fig. 4. Red = crystal-like structures, gray = amorphous matrix. **b** Mask used for the calculation of the volume fraction of these nanocrystal clusters. **c** Energy-loss spectrum from a 8 nm thickness region of $Fe_{50}Ni_{30}P_{13}C_7$ BMG. $I_0$ = number of electrons having lost no energy (the zero-loss peak), $I_t$ = total number of electrons in the EELS, $t$ = average thickness of the analyzed region, $\lambda$ = characteristic mean free path for the inelastic scattering of the material. **d** Filtered HRTEM image after autocorrelation analysis, insets 1–4 display the change in FFT patterns. **e** X-ray diffraction reveals broad diffraction maxima with no detectable crystalline peaks (scale bars 2 nm). The spatial distribution of nanocrystal clusters retrieved by autocorrelation analysis of HRTEM images from different regions, and volumetric nanocrystal cluster fraction for each region in dimensions of 21 nm × 21 nm. **f** The dispersion of nanocrystal clusters imaged from the bright zones in LM STEM. **g** The dispersion of nanocrystal clusters imaged from the relatively darker zones in LM STEM. The average volumetric fraction of the nanocrystal cluster regions (marked in pink) in (**f**) and (**g**) are 19.6 ± 1.1 μm and 13.9 ± 3.1 μm, respectively

does not strictly suit to the guiding rules for BMGs with acceptable or high GFA[50]. Since the driving force for nanocrystal formation competes with the fully amorphous glass formation upon quenching, increasing the GFA of the alloy via compositional changes or mold design would decrease the number of crystals formed, which could potentially decrease the overall extensive plasticity. A recent study shows that the cooling rate of a 1 mm diameter metallic glass rod in a copper-mold from its molten state is calculated to be 15,800 K/s[51]. This finding also highlight the fact that the critical casting rate is only size-dependent and independent of the glassy alloy composition used. The thermal gradients can be negligible for such small rod diameters (as compared to 5-cm diameter copper mold). However, the critical casting rate for these Fe-based alloy systems can be estimated to be between 10,000 and 20,000 K/s (according to ref. [50]), which means the nanocrystallization might occur upon quenching of the glassy melt. This is also one of the reasons why the $\Delta T$ and $T_{rg}$ are lower compared to that

of the many other BMG types showing higher GFA[52]. Rather, the constituent metal atoms can replace each other, and have a similar probability to occupy the lattice sites to form a solid solution, giving rise to nanocrystal formation[53,54].

In fact, Turnbull has already shown that the nucleation frequency/volume of the nanocrystals can go up to $10^{26}/m^{-3}$ as the reduced glass transition temperature $T_{rg}$ becomes 0.3–0.4[55]. It is also highlighted in the same paper that the transport of heat from the interface causes a marked recalescence within the liquid, which in turn causes homogenous nucleation and crystallization with a single nucleus. There are numerous examples of BMGs with second phase of 20% or above in volume fraction formed with similar crystallization kinetics upon fast quenching from the liquid state[11,56,57]. On the other hand, Fe-based BMGs are subjected to structural relaxation, which potentially lead to phase separation (and crystallization) even at sub-glass transition annealing[48]. It has been found that the embrittlement temperature decreases as the amount of phosphorus in the Fe-based BMG

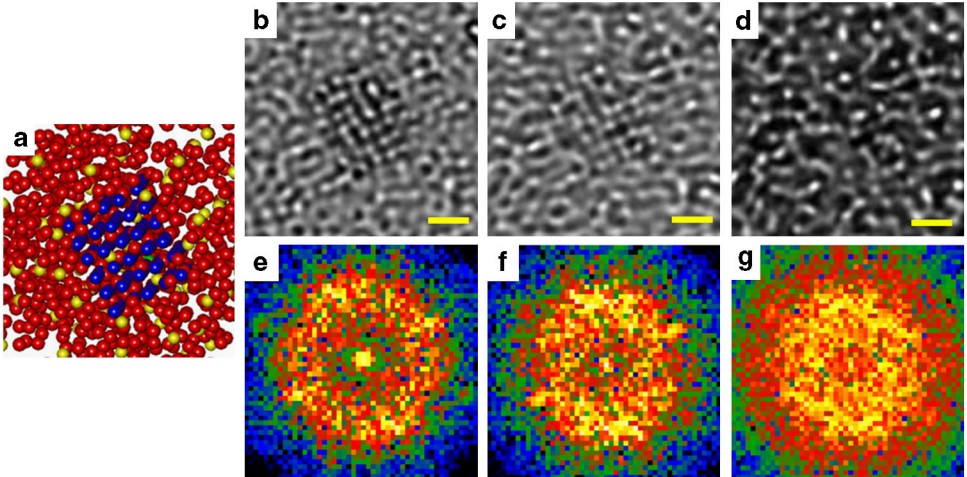

**Fig. 6** HRTEM simulations of nanocrystals observed in Fe-based BMG. **a** Atom model and **b–d** corresponding HRTEM MD simulations for the 8-, 15- and 23-nm-thick samples, respectively. **e–g** FFTs obtained from the HRTEM simulations for the 8-, 15- and 23-nm-thick samples, respectively, which show close correlation with the FFT pattern for regions with NCs in Fig. 4 (scale bars 0.5 nm)

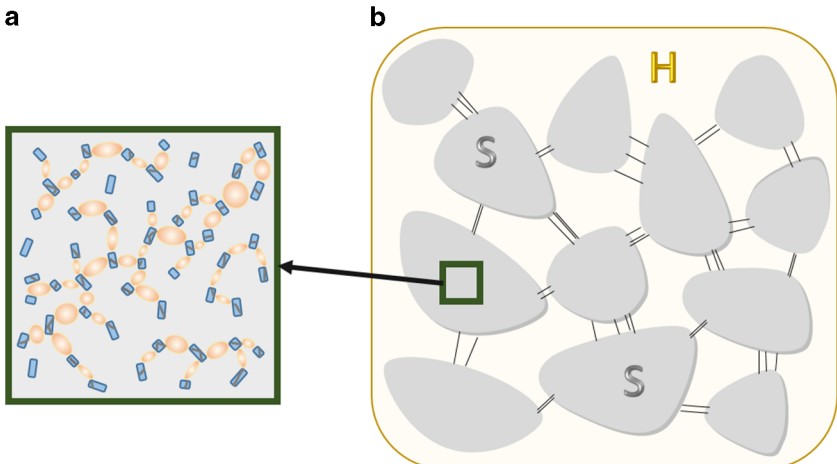

**Fig. 7** Schematics describing multiscale deformation behavior in $Fe_{50}Ni_{30}P_{13}C_7$ BMG. **a** Nanocrystals (blue particles) have lower shear modulus compared to the BMG, thereby deforming first (small gray lines within NCs). Deformation proceeds by the formation of homogenously dispersed STZs between NCs. **b** On the micro-scale, SBs do not develop into cracks because the distance between the soft regions is smaller than the critical crack length of the BMG (as in the case for ref. [9] containing plastically soft and hard regions)

increases. The embrittlement is thus caused by the segregation of phosphorus in separate regions that are less ductile than the matrix and are acting as nucleation sites[58]. However, since the driving force for crystallization during annealing is higher than that for the crystallization upon quenching, the phase separation and nanocrystallization happens quite fast and the sizes of these regions are one to two orders of magnitude larger compared to the nanocrystals formed upon quenching in our case ($1 - 1.5$ nm)[59,60]. Hence, the structural relaxation and phase separation dominates over the ductile FCC crystals formed during quenching, which probably leads to the early failure in the annealed BMGs.

For the Fe-based BMGs, previous studies suggest that in order to reach a lower energy and stability, nanocrystalline precipitates of the high temperature $\gamma - FeNi(fcc)$ phase start to form first from the glass-forming melt at elevated temperatures[61]. It is noteworthy to state that the negligible phosphorus solubility in FeNi gives rise to ejection of P to the surrounding clusters[62]. Preferential crystallization on the nano-scale was also observed in other Fe-based[63,64] and Pd-based BMGs[65]. Besides, for the Finemet-based alloys, it is also known that replacing of Ni

elements with Fe (up to 40%) leads to the retaining of the high temperature FCC-like FeNi short range atomic arrangement within the amorphous clusters[66]. These "softer" metal−metal pairs in the NCs favor the generation and propagation of SBs under compression, which is predicted to be the main mechanism for the extensive plasticity observed in the considered Fe-based BMG.

The origin of severe plasticity in Fe-BMGs is schematically explained in Fig. 7. Shear modulus of the FCC−FeNi (111) crystal, $G_{NC} = 50$ GPa is smaller than that of the composite structure ($G_{NC + BMG} = 60.5$ GPa)[51]. This finding is in agreement with the literature showing relatively lower shear moduli for the FeP(C)-based BMG alloys[32]. Poisson's ratio of the sample is measured to be $v = 0.405$, which is significantly higher compared to that for the other Fe-based BMG alloys, but also similar to the findings for the $Fe_{80}P_{13}C_7$ base alloy with no Ni addition[1]. This unexpectedly high Poisson's ratio and relatively lower shear moduli means that the energy barrier to initiate shear flow is low, which gives rise to dissipation of the store elastic energy into multiple SBs. The proliferation of SBs yields a large plastic zone size of 20 μm before cavitation is reached[5]. When the average

distance between the regions containing NCs (soft zones) is smaller than the plastic zone size (in other words, the critical crack length), the elastic energy release is smaller than the energy barrier to form an SB[15,52]. In our case, this distance is an order of magnitude lower than the estimated plastic zone size, which also explains the reason why this material shows very high fracture toughness as compared to its Fe-based BMG counterparts. Thus, localized shear (e.g. dislocations within the NCs) is primarily taken place in the NCs (homogenously dispersed blue particles in Fig. 7a). Pauly et al. showed that B2 CuZr nanocrystals in sizes of 2–5 nm homogenously dispersed within the BMG deforms by partial dislocations and twinning mechanism[12]. In fact, from our recent calculations, when the nanocrystals are below 6.7 nm for the FCC FeNi, the critical shear stress to nucleate partial dislocations is more favorable compared to perfect dislocations[67].

The deformation mechanism in metallic glasses is generally linked with the local rearrangement on the atomic level described by the generation of fertile shear transformation zones (STZ)[68,69]. The size of an STZ zone is estimated as ~2 nm (see Experimental section), corresponding to the average size of the NCs and spacing between them. Therefore, it can be postulated that the NCs (and/or their interfaces) act both as source and sink for the STZs (oval regions). This means that the NC deformation initiates STZ with similar atomic volume, which is subsequently hindered by the homogenously distributed NCs. Hence, the absorbed deformation can be widely spread through the regions with the NC particles (gray zones in Fig. 7b). Due to the plastic zone size of 20 μm that is relatively larger than the average distance between soft zones (~1 μm measured from Fig. 3a)[5], multiple SBs are expected to form (thin black lines in Fig. 7b), which in turn prevent the full development of a single SB and its propagation. Hence, the proposed multiscale deformation mechanism is believed to be the main reason behind the pronounced plasticity observed in the Fe$_{50}$Ni$_{30}$P$_{13}$C$_7$ BMG.

In this study, we provide evidence for the presence of metallic nm-sized crystals homogenously dispersed within the soft regions of the Fe-based BMG using aberration-corrected HRTEM. Since Fe and Ni atoms have a very small heat of mixing and similar atomic radii, the GFA of this system decreases as the Ni content increases, which gives rise to nanocrystal formation. Remarkable variations in the mechanical properties measured by the nanoindentation tests and preferential thinning confirm the existence of hard and soft phases for this BMG. TEM simulations correlate well with the experimental findings, where the identification of the nanocrystalline precipitates is only possible below a threshold lamellae thickness of 20 nm. In consideration of these findings, one can draw the conclusion that these nanocrystals contribute to the remarkable shear observed under quasi-static compression and enhance the plasticity of the BMG through the initiation and impediment of STZs. The controlled nanocrystal-induced plasticity with crystal sizes of a several nanometers can help better to understand the deformation mechanism in other BMG classes exhibiting extensive plasticity.

## Methods

**Sample preparation**. Fe$_{50}$Ni$_{30}$P$_{13}$C$_7$ BMG master alloy ingots were prepared from elements with purity higher than 99.99% using an Edmund Bühler GmbH MAM-1 Compact arc melter. The master alloys were subsequently heated above the liquidus temperature five times to form a homogenous mixture. The master alloys were then crushed into 0.5 g pieces. The molten liquid glass did not even start flowing into the 1 mm diameter copper mold without fluxing. Therefore, to purify the BMG ingot and improve the GFA of the FeNi-based BMGs[5,29,34], the fluxing technique was used by exposing the BMG ingot to B$_2$O$_3$ fluxing agent and kept at 1400 K for 9 h. Rods with 1 mm diameter and 3 cm length were cast in a water-cooled copper mold (>1000 K/s). The casting was conducted under Ar atmosphere using an in situ suction casting device attached to the same arc melter.

**Sample characterization**. For rods with up to 1.8 mm diameter a fully amorphous structure was observed for Fe$_{50}$Ni$_{30}$P$_{13}$C$_7$ BMG as confirmed by the DSC and XRD measurements[34]. The rods with 1 mm diameter were then sliced, ground, and polished with diamond paste to a thickness of 60 μm. Subsequently, the samples were exposed to double-sided ion milling (Gatan model 691) using argon ions with a primary beam energy of 4 keV at a rotation speed of 2.5 rpm. For the first 20 min, the gun angles were set to 6° for 20 min, followed by 4° for 70 min. In order to make sure that no nanocrystals are formed during ion milling, continuous liquid nitrogen cooling was applied. The temperature of the sample during the ion milling was kept around −60 °C. HRTEM studies were performed on Cs-corrected Titan G2 60–300 TEM (FEI, Netherlands) equipped with a high brightness gun, monochromator and Gatan Quantum GIF (Gatan, USA). Imaging and EELS spectroscopy was performed at 300 kV with a resolution of about 0.1 nm in the HRTEM mode to facilitate atomic resolution imaging. No electron damage was observed on the sample at these specific imaging conditions during image acquisition time. HRTEM image simulations were performed using MUSLI simulation package[70] and a set of custom Digital Micrograph scripts. Propagation was calculated on 512 × 512 K matrix (0.05 nm per pixel) with 0.05 nm slice thickness. Coherent aberrations up to fifth order were accounted for as well as incoherent aberrations due to chromatic aberration and instability of objective lens. Dose statistics corresponding to experiment was then superimposed on the images and modulation transfer function of CCD camera was applied. Such approach was confirmed to eliminate a Stobbs factor from simulated images and was proven to provide quantitative contrast. Because the nanocrystal sizes are 1.0–1.5 nm inside much thicker and larger amorphous matrix of the same elements, EDX and EELS analysis were extremely challenging to chemically resolve their content. Mechanical tests were conducted with the Kammrath & Weiss Tensile/Compression Module (with 5 kN load cell) with attached compression fixtures. The visualization of the deformation process, and the strain correction was implemented using a video extensometer system (MicroDAC strain measurement) attached to the testing device. The fractographic analysis after rupture was performed using an SEM Zeiss Ultra Plus. The macroscopic deformation and the correlated SB formation was investigated using Olympus BX51. The indentation tests were conducted on a Hysitron Triboscope system (now Bruker, Billerica, MA, USA), its transducer mounted on the scanner head of a Digital Instruments (now also Bruker) D3100. The indenter tip in use was a diamond Berkovich indenter, its area function (projected contact area as a function of contact depth) calibrated by standard calibrations on a fused silica sample. The indentations were performed in load control mode. Starting with a linear ramp up to the maximum load of 10 mN within 2 s, the loading course was continued by holding the maximum load for 10 s, and finally by unloading to zero load within 2 s. The fast loading and unloading sequences were chosen in order to minimize the influence of thermal instrumental drift, the duration at maximum load in order to minimize the influence of creep during unloading. Hardness and elastic modulus were determined using the well-established method proposed by Oliver and Pharr[71]. The elastic modulus values are given in terms of the reduced indentation modulus that is affected by the elastic deformation of the indenter tip as well as by the lateral elastic extension of sample contact area and of the tip and can, therefore, not immediately be compared with the Young's modulus obtained from the universal compression testing. However, variations of the elastic modulus can still be obtained.

The LAMMPS MD simulation software was used to generate the amorphous BMG matrix, and the TEM simulations were performed using the custom DigitalMicrograph scripts. The algorithm gives a maximum intensity at the position of the crystal-like structure and vice versa: visually irregular structure corresponds to small values of the estimator. Thus, detecting a nanocrystal in the amorphous matrix by HRTEM just depends on the sample thickness that should be smaller than 20 nm, and the density of the nanocrystal in amorphous matrix. The Poisson's ratio, $v$, of the Fe$_{50}$Ni$_{30}$P$_{13}$C$_7$ BMG obtained from the ultrasonic measurements as 0.405, which is measured using an Olympus 5900PR ultrasonic pulse-receiver, correlates well with other BMGs showing high plasticity[3,4]. For the sample preparation, the cast rod of 40 mm in length was laterally glued to a plate, and both sides were equally grinded and subsequently polished with the paper of 4000 grit size. Final thickness of the sample was 0.6 mm. Measurements were taken from three different points and averaged. The standard deviation of the measurement was found as $v = 0.405 \pm 0.05$. This finding confirms the accuracy in measuring the velocities using ultrasonic pulses, which is approximately equal to 0.5% for shear waves and 1% for longitudinal waves. Similarly, the density was measured as $\rho = 6.2 \pm 0.1$ g/cm$^3$ using the Archimedes method from a 100 mg bulk rod of 1 mm diameter. Typical magnetization of pure Fe and Ni are 217.6 and 55.1 emu/g[72]. Our preliminary findings from the magnetic measurements performed by a SQUID magnetometer show 120 emu/g, which corroborates the decrease in magnetization upon Ni addition. During the measurement, we did not encounter any jamming or loss in the signal.

**Theoretical calculations**. Based on the Johnson−Samwer cooperative shear model of STZs[73], using the equations and parameters given in refs. [73,74], and current findings from the nanoindentation measurement (e.g. hardness average, $H_{avg} = 6840.56$ GPa calculated from the ratio $P_{max}$ to the projected contact area, $A$), and ultrasonic measurements ($E = 170 \mp 2$ GPa, yielding $G = 60.5 \mp 2$ GPa), the strain rate sensitivity measurement obtained from nanoindentation for this Fe-BMG $m =$

0.337, the STZ volume of the corresponding BMG (by direct differentiation of the activation barrier energy for shear) was estimated at 8.23 nm$^3$. This corresponds to approximately 642 atoms (calculated from the statistical averaging of the constituent elements[74] present within an STZ. Thus, the estimated STZ zone size for a representative cubic deformation region is 2.02 nm. The critical crystal size below which the activation of partial dislocations are more favored compared to perfect dislocations is found from the formula:

$$D_c = \frac{2\alpha\mu(b_N - b_P)b_P}{\gamma},\qquad(2)$$

where $\alpha = 1$ (mixture of edge and screw dislocations), $\mu = 50$ GPa (shear modulus of FeNi FCC crystal[51]), and $\gamma = 79$ mJ/m$^2$ (stacking fault energy of FeNi FCC crystal[75]). Burgers vector of the perfect dislocation, $b_N$, and Shockley partial dislocation, $b_P$, are calculated from the equations:

$$\| b_N \| = (a/2)\sqrt{h^2 + k^2 + l^2},\qquad(3)$$

where $a = 2.08$ Å (unit cell size), and the dislocation takes place on the [110] plane.

$$\| b_P \| = \left(\frac{a}{6}\right)\sqrt{h^2 + k^2 + l^2},\qquad(4)$$

and the dislocation takes place on the [112] plane. Hence, the critical crystal size, $D_c$, is found to be 6.70 nm.

## Data availability

All data are available upon request from the corresponding or lead authors.

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

## Acknowledgements

The authors thank Michael Frey for the preparation of the alloys, Gabriele Felber for the TEM preparation of the samples, Amir Rezvan for preparing the DSC sample, and Christoph Gammer for the fruitful discussions about the TEM studies. This work was supported by the European Research Council under the Advanced Grant "INTELHYB—Next Generation of Complex Metallic Materials in Intelligent Hybrid Structures" (Grant ERC-2013-ADG-340025) and the Austrian Science Fund (FWF): No. P29148-N36.

## Author contributions

B.S and Y.P.I. conceived the project. Y.P.I. performed the TEM investigations and simulations. M.S. produced the alloys by suction casting. B.S. performed the quasi-static compression tests. T.S. and B.S. performed the nanoindentation tests. B.S. and Y.P.I. performed data analysis and wrote the manuscript. A.C. and Z.Z contributed to the discussions and manuscript writing. J.E. supervised the project, and contributed to the enhancement of the manuscript.

## Additional information

**Competing interests:** The authors declare no competing interests.

