## [Peer Review File(PDF 1209 kb) · Nature Communications]

Reviewers' Comments:

Reviewer #1:

Remarks to the Author:

In this manuscript, the authors demonstrate the origin of large plasticity and multiscale effects in the ductile Fe₅₀Ni₃₀P₁₃C₇ metallic glass by using aberration-corrected high-resolution transmission electron microscopy (HRTEM) and HRTEM MD simulations. Nanocrystals on the order of 1-1.5 nm in size are observed and the accumulation of nanocrystals is linked to the hard and soft zones. Moreover, the formation of softer regions is considered originating from the homogeneously dispersed nanocrystals, which are responsible for the start and stop mechanism of shear transformation zones, and hence, playing the key role in the enhancement of plasticity. The reviewer believe that this work is interesting and worth publishing in Nature Communication. However, there are a few issues needed to be addressed before acceptance.

1. Generally, it is difficult to measure the elastic constant of magnetic Fe-based bulk metallic glasses accurately using ultrasonic measurements. How is the elastic constant measured in this work? This part is contradictory, the authors should clearly explain what is the starting process of the sample they used for this analysis and should mention the dimension of the samples.
2. The ductile Fe-based bulk metallic glasses will be brittle after annealing. The authors should try to interpret the underlying mechanism within the framework of this work.
3. How to interpret the size effect of plasticity for BMGs within the framework of this work?
4. Page 6 line 134. "Fig.1c" should be "Fig.2c"?
5. Page 9 line 190. "iron-based" should be "Fe-based" for consistency.

Reviewer #2:

Remarks to the Author:

Fe-based bulk metallic glasses (BMG) have attracted great research interest ever since their first synthesis in 1995, because of the combination of their superior chemical, magnetic and mechanical properties. Unfortunately, they usually fracture catastrophically when deformed at room temperature, e.g., undergoing only a few percent of plastic strain in compressions, which limits their widespread applications. Recently, it has been reported that a newly-developed 1-mm diameter Fe₅₀Ni₃₀P₁₃C₇ BMG possesses extensive plasticity (up to 22% plasticity) together with a yield strength of 2250MPa. However, a physical understanding of the extensive plasticity mechanisms is still unclear. In this manuscript, the authors demonstrated that the homogeneously dispersed nanocrystals play the key role in the of large plasticity in the ductile Fe₅₀Ni₃₀P₁₃C₇ metallic glass by using aberration-corrected high-resolution transmission electron microscopy (HRTEM), HRTEM MD simulations and theoretical model. This work is very interesting and important for future syntheses of tough Fe-based BMGs. Hence, this manuscript is worth to publish in Nature Communication. There are only a few issues needed to be addressed before acceptance.

- (1) How to simulate the HRTEM images in this work? I think the details should be given in the part of methods.
- (2) Maybe in Page 6 line 134. "Fig.1c" should be replaced by "Fig.2c".
- (3) The critical diameter of Fe₅₀Ni₃₀P₁₃C₇ BMG is about 1.8 mm by water quenched after fluxing. How about the critical diameter of this composition by casting using a water-cooled copper mold?
- (4) The DSC curve of Fe₅₀Ni₃₀P₁₃C₇ BMG from water-cooled copper mold casting should be added in the manuscript.

Reviewer #3:

Remarks to the Author:

The authors present a study of the nanoscale structure of a Fe-based bulk metallic glass. The Fe-based BMG shows a combination of high strength and good plasticity in compression. High resolution electron microscopy is employed to study the nanoscale structure of the BMG. HRTEM

reveals nanometer size FeNi crystals dispersed throughout the matrix. The improved plasticity is suggested to originate from a combination of high Poisson's ratio and soft regions (nc-FeNi), which facilitate proliferation of shear bands and deformation of the nc-FeNi.

While the results and proposed deformation mechanisms are intriguing, numerous questions and concerns remain as follows.

-It would be highly beneficial to include XRD patterns and DSC results for the as cast alloy. This would show that the sample is macroscopically amorphous. Also, information from the DSC trace (i.e. T_g , T_x , and excess enthalpy) are highly relevant to the later discussion of cooling rate effects.

-The nanoindentation results require additional explanation of the methods and results. How are the indentation curves analyzed? While the compressive strength is measure to be 2265 MPa, the hardness is reported as 28.5 to 36.5 GPa. Such high hardness is comparable to sapphire and suggests a Tabor factor greater than 10. Is there an explanation for why this BMG alloy exhibits such high hardness and Tabor factor? Showing indentation loading curves would make such extreme hardness easier to understand.

In addition the comparison of the spatial variation observed by STEM and nanoindentation are unconvincing to due to the differences in length scales, especially in the absence of nanoindentation curves to illustrate the variations in depth and volume interrogated.

-The discussion of cooling rate effects on pg. 5, lines 108-114 might be better placed in the discussion as it seems rather speculative rather than presenting a measured result with a corresponding explanation.

-Additional illustration of the process for identifying nanocrystals via the autocorrelation method is needed. For instance, does a selected area diffraction pattern reveal distinct diffraction spots for the region shown in Fig. 4a? Does the FFT of the image in Fig4a show the same spots? Can a comment on the sensitivity of the method be made, especially since the imaging simulation results in Fig. 5 indicate detecting a nanocrystal in the amorphous matrix is challenging?

Furthermore, could elemental mapping through a combination of EELS and EDS improve identification of the nanocrystals? The discussion suggests that the metalloids (C and P) preferentially pair with Fe and that the FeNi clusters eject phosphorus. Shouldn't this produce phase separation?

Point-by-point response sheet

We would like to thank the referees for their insightful comments. We went through the questions carefully, and tried to address all the points raised by the referees. In the revised manuscript, the editor and referees can find the changes highlighted.

Remarks to the editor:

Q: Your manuscript entitled "Origin of large plasticity and multiscale effects in Fe-based metallic glasses" has now been seen by 3 referees. You will see from their comments below that while they find your work of interest, some important points are raised. These include but are not limited to experimental clarifications and added discussion. We are interested in the possibility of publishing your study in Nature Communications, but would like to consider your response to these concerns in the form of a revised manuscript before we make a final decision on publication.

We would like you to explicitly discuss how the current work differs from reference 7, which is a concern that was highlighted in the remarks to the Editor.

R: We would like to thank the referees for finding our work interesting and worth to be published in Nature Communications, and express our gratitude to the editor who is interested in the possibility of publishing our article in Nature Communications after the necessary changes.

The main goal of this paper is to resolve, for the first time, the origin of large plasticity from an atomic- and micro-scale point of view using a combination of an advanced TEM and nanoindentation techniques, which bring in novelty and high impact to this research field. The extent of plasticity in monolithic glasses is linked with the existence of homogeneously dispersed nanocrystals formed during fast quenching rates which are responsible for the initiation and impediment of the plastic deformation. Aberration-corrected HRTEM technique used in this study has the point-to-point resolution less than 0.1 nm, which allows to acquire easily an image of nanocrystal in sizes of 1-1.5 nm of the very thin samples. Therefore, the findings of this study give deeper insight to Yang *et al.*'s paper on the unprecedented plasticity observed for the considered Fe-based BMG. In addition, this paper aims for an understanding of the role of the extremely small crystals and their nano- and micro-scale dispersion on the overall large deformation, which brings a completely new explanation to the deformation mechanism in other BMG classes exhibiting extensive plasticity. To highlight the importance of this work, we have included a similar text into the Introduction section.

Referee : 1

Comment: In this manuscript, the authors demonstrate the origin of large plasticity and multiscale effects in the ductile Fe₅₀Ni₃₀P₁₃C₇ metallic glass by using aberration-corrected high-resolution transmission electron microscopy (HRTEM) and HRTEM MD simulations. Nanocrystals on the order of 1-1.5 nm in size are observed and the accumulation of nanocrystals

is linked to the hard and soft zones. Moreover, the formation of softer regions is considered originating from the homogeneously dispersed nanocrystals, which are responsible for the start and stop mechanism of shear transformation zones, and hence, playing the key role in the enhancement of plasticity. The reviewer believe that this work is interesting and worth publishing in Nature Communication. However, there are a few issues needed to be addressed before acceptance.

R: We would like to thank referee 1 for finding our work interesting and support for publication our article in Nature Communications. We considered carefully all the technical points raised by the referee.

Q1: Generally, it is difficult to measure the elastic constant of magnetic Fe-based bulk metallic glasses accurately using ultrasonic measurements. How is the elastic constant measured in this work? This part is contradictory, the authors should clearly explain what is the starting process of the sample they used for this analysis and should mention the dimension of the samples.

R1: We thank the referee for giving us a chance to elaborate our findings on elastic constants. In the revised version, we have included the details of sample preparation and ultrasonic measurement. “For the sample preparation, the cast rod of 40 mm in length was laterally glued to a plate, and both sides are equally grinded and subsequently polished with the paper of 4000 grit size. Final thickness of the sample was 0.6 mm. Measurements were taken from 3 different points and averaged. The standard deviation of the measurement is found as $\nu = 0.405 \pm 0.05$. This finding confirm the Wang et al.’s review paper, the accuracy in measuring the velocities is approximately equal to 0.5% for shear waves and 1% for longitudinal waves. Similarly, the density was measured as $\rho = 6.2 \pm 0.1 \text{ g/cm}^3$ using the Archimedes method from a 100 mg bulk rod of 1mm diameter.” We have added the corresponding text to the Methods section.

In addition, we have also added the following text to the Methods section: “Typical magnetization of pure Fe and Ni are 217.6 and 55.1 emu/g [Crangle_The Magnetization of Pure Iron and Nickel, Proc. R. Soc. Lond. A, 321, 477-491 (1971)]. Our preliminary findings from the magnetic measurements performed by a SQUID magnetometer show 120 emu/g, which corroborates the decrease in magnetization upon Ni addition. During the measurement, we did not encounter any jamming or loss in the signal.”

Due to the large plasticity observed, FePC-based BMGs have a tendency to exhibit relatively higher Poisson’s ratio as compared to other Fe-based BMGs of interest (0.4 confirmed for Fe80P13C7 based BMGs) [J. J. Lewandowski, Phil. Mag. Lett. 85, 2 (2005) & L.A. Davis, in Metallic Glasses (ASM, Metals Park, Ohio, 1978), pp. 191–223] This previous finding points out that the Poisson’s ratio for Fe50Ni30P13C7 showing much higher plasticity can be slightly higher ($\nu=0.405$). Moreover, almost all the literature findings for the elastic constants of the Fe-based magnetic BMGs rely on the ultrasonic measurements [Lewandowski_ Tough Fe-based bulk metallic glasses, Appl. Phys. Lett., 92, 091918 (2008), Gu_Mechanical properties, glass transition temperature, and bond enthalpy trends of high metalloid Fe-based bulk metallic glasses, Appl. Phys. Lett., 92, 161910 (2008); Wang_Correlations between elastic moduli and properties in bulk metallic glasses, J. Appl. Phys., 99, 093506 (2006)].

Q2: The ductile Fe-based bulk metallic glasses will be brittle after annealing. The authors should try to interpret the underlying mechanism within the framework of this work.

R2: We certainly agree with the referee that the ductile Fe-based bulk metallic glasses will be subjected to structural relaxation, which potentially lead to phase separation (and crystallization) even at sub-glass transition annealing [Yao_Fe-based bulk metallic glass with high plasticity, *Appl. Phys. Lett.*, 90, 061901 (2007)]. It has been also found that the embrittlement temperature decreases as the amount of phosphorus in the Fe-based BMG increases. The embrittlement is thus caused by the segregation of phosphorus in separate regions which are less ductile than the matrix and are acting as nucleation sites. [Walter_The Ductile-Brittle Transition of Some Amorphous Alloys_Mater. Sci. Eng. 33, 91 (1978)]. However, since the driving force for crystallization during annealing is higher than that for the crystallization upon quenching, the phase separation and nanocrystallization happens quite fast and the sizes of these regions are one to two orders of magnitude larger compared to the nanocrystals formed upon quenching in our case (1 - 1.5 nm) [Kharkov_The Relation Of High-Temperature Stability Of Amorphous Alloys With Nucleation And Growth Of The Crystalline Phases, *J. Non-Cryst. Solids*, 117/118 244 (1990); Duarte_Kinetics and crystallization path of a Fe-based metallic glass alloy, *Acta.Mater.*, 127, 341 (2017)]. Hence, the structural relaxation and phase separation dominates over the ductile FCC crystals formed during quenching, which probably leads to the early failure in annealed BMGs. A relevant text is inserted to the Discussion section in the revised version.

Q3: How to interpret the size effect of plasticity for BMGs within the framework of this work?

R3: As we have previously highlighted in the Discussion section, when the average distance between the regions containing nanocrystals (soft zones) is smaller than the plastic zone size (in other words, the critical crack length), the elastic energy release is smaller than the energy barrier to form a shear band. [Volkert_Effect of sample size on deformation in amorphous metals, *J. Appl. Phys.*, 103, 083539 (2008); Sarac_Designing tensile ductility in metallic glasses, *Nat.Comm.* 4:2158 (2013)]. In our case, this distance is maximum several microns, which is an order of magnitude lower than the estimated plastic zone size of 20 μm . An extended comparison between shear bands and distance between hard/soft zones is provided in the revised version. Moreover, as we previously indicated in the schematics (Fig 6a), extensive plasticity is achieved by the multiple nucleation and stop mechanism of shear transformation zones (STZs) between the densely populated nanocrystals in sizes of 1 to 1.5 nm. Hence, the distribution of the microscale soft regions and the existence of nanoscale crystals therein mainly determines the extent of deformation of BMGs.

Increase of the overall size of the specimen is only possible by increasing the glass forming ability through compositional change. However, since the driving force for nanocrystal formation competes with the fully amorphous glass formation upon quenching, increasing the diameter of the rod will decrease the number of crystals formed, which could potentially decrease the overall extensive plasticity. A relevant passage is also inserted into the Discussion section for a better clarification of the size effect.

Q4. Page 6 line 134. “Fig.1c” should be “Fig.2c”?

R4: We thank the reviewer for this correction, the change is applied accordingly.

Q5. Page 9 line 190. “iron-based” should be “Fe-based” for consistency.

R5: For consistency, we replaced the term “iron-based” with “Fe-based”.

Referee : 2

Comment: Fe-based bulk metallic glasses (BMG) have attracted great research interest ever since their first synthesis in 1995, because of the combination of their superior chemical, magnetic and mechanical properties. Unfortunately, they usually fracture catastrophically when deformed at room temperature, e.g., undergoing only a few percent of plastic strain in compressions, which limits their widespread applications. Recently, it has been reported that a newly-developed 1-mm diameter Fe₅₀Ni₃₀P₁₃C₇ BMG possesses extensive plasticity (up to 22% plasticity) together with a yield strength of 2250MPa. However, a physical understanding of the extensive plasticity mechanisms is still unclear. In this manuscript, the authors demonstrated that the homogeneously dispersed nanocrystals play the key role in the of large plasticity in the ductile Fe₅₀Ni₃₀P₁₃C₇ metallic glass by using aberration-corrected high-resolution transmission electron microscopy (HRTEM), HRTEM MD simulations and theoretical model. This work is very interesting and important for future syntheses of tough Fe-based BMGs. Hence, this manuscript is worth to publish in Nature Communication. There are only a few issues needed to be addressed before acceptance.

R: We express our gratitude to the second reviewer for his/her general description of bulk metallic glasses, and identifying the fact that the physical understanding behind the extensive plasticity of the newly developed Fe₅₀Ni₃₀P₁₃C₇ BMG became clear through the high-resolution transmission electron microscopy (HRTEM), MD simulations and theoretical model presented in our study, which will be essential to develop tough Fe-based BMGs in the future. In the revised version, we considered carefully all the technical points raised by the referee.

Q1. How to simulate the HRTEM images in this work? I think the details should be given in the part of methods.

R1: In the revised version, the referee can find the details of the HRTEM simulation within the Methods section (pasted below).

“HRTEM image simulations were performed using MUSLI simulation package and a set of custom Digital Micrograph scripts. Propagation was calculated on 512x512K matrix (0.05 nm/pixel) with 0.05 nm slice thickness. Coherent aberrations up to 5th order were accounted for as well as incoherent aberrations due to chromatic aberration and instability of objective lens. Dose statistics corresponding to experiment was then superimposed on the images and modulation transfer function of CCD camera was applied. Such approach was confirmed to eliminate a Stobbs factor from simulated images and was proved to provide quantitative contrast.”

Q2. Maybe in Page 6 line 134. “Fig.1c” should be replaced by “Fig.2c”.

R2: We apologize for this typo, the term “Fig.1c” is replaced by “Fig.2c”.

Q3. The critical diameter of Fe₅₀Ni₃₀P₁₃C₇ BMG is about 1.8 mm by water quenched after fluxing. How about the critical diameter of this composition by casting using a water-cooled copper mold?

R3: To obtain a higher glass forming ability and to be consistent with the literature, we followed the already developed fluxing technique for the same and similar Fe-based BMGs followed by water quenching. “The molten liquid glass did not even start flowing into the 1 mm diameter copper mold without fluxing.” We have added this last sentence to the Methods section to show the importance of fluxing in casting Fe₅₀Ni₃₀P₁₃C₇ BMG.

Q4. The DSC curve of Fe₅₀Ni₃₀P₁₃C₇ BMG from water-cooled copper mold casting should be added in the manuscript.

R4: Following the suggestion of the referee 2, DSC curve for the 1 mm water-cooled copper mold casting of the Fe₅₀Ni₃₀P₁₃C₇ BMG and explanation is provided in the revised version of the manuscript. The glass transition T_g , crystallization temperature T_x , and heat of crystallization ΔH_x is provided in the main figure, whereas the melting temperature T_m , liquidus temperature T_l , curie temperature T_c , and heat of melting ΔH_m is provided as an inset.

Referee : 3

Comment: The authors present a study of the nanoscale structure of a Fe-based bulk metallic glass. The Fe-based BMG shows a combination of high strength and good plasticity in compression. High resolution electron microscopy is employed to study the nanoscale structure of the BMG. HRTEM reveals nanometer size FeNi crystals dispersed throughout the matrix.

The improved plasticity is suggested to originate from a combination of high Poisson's ratio and soft regions (nc-FeNi), which facilitate proliferation of shear bands and deformation of the nc-FeNi. While the results and proposed deformation mechanisms are intriguing, numerous questions and concerns remain as follows.

R: We thank referee 3 for his insightful comments and his constructive attitude towards our work. In the revised version, we tried to address all the points one-by-one in a detailed manner.

Q1: It would be highly beneficial to include XRD patterns and DSC results for the as cast alloy. This would show that the sample is macroscopically amorphous. Also, information from the DSC trace (i.e. T_g, T_x, and excess enthalpy) are highly relevant to the later discussion of cooling rate effects.

R1: We thank the referee for this suggestion; in the revised version we have added the XRD pattern (Fig. 5e) and DSC curve (Fig. 2) of the 1 mm diameter fluxed and copper mold quenched Fe₅₀Ni₃₀P₁₃C₇ BMG. Further discussion pertaining to the fully amorphous nature, as well as the extent of the supercooled liquid region and crystallization enthalpy is provided in the revised manuscript.

Q2: The nanoindentation results require additional explanation of the methods and results. How are the indentation curves analyzed? While the compressive strength is measure to be 2265 MPa, the hardness is reported as 28.5 to 36.5 GPa. Such high hardness is comparable to sapphire and suggests a Tabor factor greater than 10. Is there an explanation for why this BMG alloy exhibits such high hardness and Tabor factor? Showing indentation loading curves would make such extreme hardness easier to understand.

In addition, the comparison of the spatial variation observed by STEM and nanoindentation are unconvincing to due to the differences in length scales, especially in the absence of nanoindentation curves to illustrate the variations in depth and volume interrogated.

R2: The hardness values we have reported in the original version were definitely wrong. Thank you for adverting to this point. Unfortunately, we have used for evaluating the contact area the area function (projected contact area as a function of the contact depth) of our cube corner indenter instead of the Berkovich indenter, the latter definitely having been used for the present work. We deeply apologize for the troubles caused by this error. The reasons were mainly two: First, during the time of our measurements we performed another set of experiments on the same sample, but with the other indenter (cube corner), rather simultaneously. Second, we had a lot of troubles with the measuring computer in use, finally ending in a final crash even of the hard disk. Thus, we checked, after having read your totally justified criticism, which indenter had been definitely used for the present investigation. For this purpose, we performed some further indentations on the original sample with either indenter tips and compared the resulting curves and contact depths with those of the original work. This gave us clear evidence that for the present work the Berkovich tip had been used. In addition, it showed that the original curves and the contact depths had fortunately been correct. From these contact depths and the knowledge of the original area function (that had been stored additionally on an office computer) we then recalculated the individual contact areas and consequently hardness and modulus. All the calculations we performed applying the standard, well-established Oliver-

Pharr method, see reference 78 in the revised manuscript. The hardness values now vary between 6.15 and 8.57 GPa, resulting in an average Tabor factor close to 3, as stated now in the revised manuscript. Due to your suggestion, we added a figure showing the load-displacement curves as Fig.3c. Based on this new hardness average, we have also modified the calculated STZ volume as well as the number of atoms present within the STZ in the Methods-Theoretical Calculations. Moreover, we estimated the lateral plastic zone extension, varying between 1.5 and 4 μm , depending on which of the common models is used. Additional explanation of the methods is now given in the last chapter, Methods, due to your suggestion.

Q3: The discussion of cooling rate effects on pg. 5, lines 108-114 might be better placed in the discussion as it seems rather speculative rather than presenting a measured result with a corresponding explanation.

R3: We thank the referee for this suggestion; the corresponding section is now placed to the Discussion section.

Q4: Additional illustration of the process for identifying nanocrystals via the autocorrelation method is needed. For instance, does a selected area diffraction pattern reveal distinct

diffraction spots for the region shown in Fig. 4a? Does the FFT of the image in Fig4a show the same spots? Can a comment on the sensitivity of the method be made, especially since the imaging simulation results in Fig. 5 indicate detecting a nanocrystal in the amorphous matrix is challenging?

Furthermore, could elemental mapping through a combination of EELS and EDS improve identification of the nanocrystals? The discussion suggests that the metalloids (C and P) preferentially pair with Fe and that the FeNi clusters eject phosphorus. Shouldn't this produce phase separation?

R4: A detailed description of the autocorrelation procedure is provided in ref. 49, where corresponding Digital Micrograph scripts can be downloaded as well. Image of Fig. 5a is in fact similar to Fig4a,f, i.e. FFT of the red regions on Fig5a show diffraction spots while grey regions do not. SAED analysis area is typically at least 100nm, which would cover multiple fields of view of the image on Fig5a, so the diffraction spots from the nanoclusters are rotationally averaged and spots do not show up – experimentally we observe rotationally symmetric ring diffraction pattern which is typically interpreted as an amorphous structure.

The advantage of HRTEM imaging is the visualization of the sample structure at atomic level. Modern aberration corrected TEM (used at the present study) allows to image the thin samples (below 20 nm thickness) with the spatial resolution below 0.07 nm. So the nanocrystals with the size from 1.0 nm are clearly visible. Our TEM simulations also support the appearance of the lattice fringes from 1 nm nanocrystal in amorphous matrix with the thickness up to 20 nm (in general it will also depend on the material of the sample). In the present manuscript we used an approach based on the HRTEM imaging to have a direct access to such small nanocrystals in amorphous matrix. This section is placed into the Introduction section for better comprehension of the technique adopted in this study.

Figure shows the filtered HRTEM image (for better visualization of the crystal-like areas) after autocorrelation analysis (Fig.4a in the manuscript). In inserts FFT patterns for the areas of different sizes are shown. FFTs from the small regions (comparable to the nanocrystal size) 1-2 show pronounced lattice fringes. The larger area depicts pronounced amorphous ring from the matrix which is strongly affected on the visualisation of the fringes from nanocrystals. Similar effect is shown in the literature by Fluctuation Electron Microscopy (nanodiffractions), only at the beam sizes comparable to the area of interest the effect of the medium range order is clearly visible. The collection of the FFTs only from the areas with nanocrystals shows more pronounced lattice fringes on the resulted FFT pattern. The relevant passage is placed to the Results section.

Discussion on sensitivity of the method is very tricky as it requires at the first place the definition of sensitivity which cannot be made in objective way – definition of any kind of threshold value will be subjective. Obviously such discussion is far beyond the scope of this paper. What can be told from observations is that once the crystal is visually recognizable – the algorithm gives a maximum at this position and vice versa: visually irregular structure corresponds to small values of the estimator. Thus, the detecting a nanocrystal in the amorphous matrix by HRTEM just depends on the sample thickness which should be smaller than 20 nm, and the density of the nanocrystal in amorphous matrix.

Atomic resolution EDX and EELS techniques are now available on dedicated TEMs. But it is possible only for the specific lattice and zone axis of the monocrystalline materials where atoms of different elements forms separated atomic columns on the STEM images. Unfortunately, it is extremely challenging to resolve chemically 1.0-1.5 nm nanocrystals inside much thicker amorphous matrix of the same elements.

Reviewers' Comments:

Reviewer #1:

Remarks to the Author:

The authors has been modified their manuscript according to the reviewers' comments in detail. The paper has been greatly improved. So it can be accepted to publish in Nature Communication

Reviewer #2:

Remarks to the Author:

Authors have revised the manuscript according to reviewer's comments, therefor the manuscript can be accepted for publication now in NC officially.

Reviewer #3:

Remarks to the Author:

Here, the authors have provided a detailed response to all comments and suggestions. Overall, the revisions are satisfactory with a few minor modifications.

-Please correct Figure 3b to reflected the corrected nanoindentation analysis.

-The text must be edited for better consistency. The discussion paragraphs on effect of specimen size (pg. 11), critical cooling rate (pg. 11-12), and embrittlement annealing (pg. 12) are import points but do not flow with the overall narrative.

-Is there any evidence of spatial variations in nanocrystal concentration? This is suggested to be the origin of softer regions but no evidence of this spatial variation is provided.

-Please proofread for minor errors. For instance, on pg. 10 "There are numerous examples of BMGs of second nano- of 20%..."

-If possible, using a consistent unit for STZ size would be helpful. The main body uses nm while the theoretical calculation methods uses nm^3 .

Point-by-point response sheet

We would like to thank all the referees for finding the revision satisfactory, and accepting the article to be published in Nature Communications. We also express our gratitude to the editor who is interested in the possibility of publishing our article in Nature Communications after the final modifications. We tried to address all the concerns raised by the referee 3 and the editorial board.

Remarks to the Editor:

The authors have edited the manuscript in accordance with the suggestions from the editorial board. We have revised the abstract and introduction section. All the subfigures were numbered, and to increase the legibility and increase the coherency, the indicated insets were turned into subfigures. We also paid attention to the first use of the abbreviations. Data availability is now provided in the Methods section. All the other changes were completed in agreement with the comments provided.

In addition, in order to comply with the reference standards while retaining the consistency of the manuscript, we have reduced down the number of citations from 89 to 75. We also approve the two-sentence editor's summary.

Remarks to the referee 3:

Comment: Here, the authors have provided a detailed response to all comments and suggestions. Overall, the revisions are satisfactory with a few minor modifications.

R: We would like to thank referee 3 for his/her insightful comments. In the final revision, we tried to address the minor concerns raised by the referee 3.

Q1: Please correct Figure 3b to reflected the corrected nanoindentation analysis

R1: We apologize for this mistake, in the final revision Figure 3b – Hardness plot is now redrawn with the corrected data.

Q2: The text must be edited for better consistency. The discussion paragraphs on effect of specimen size (pg. 11), critical cooling rate (pg. 11-12), and embrittlement annealing (pg. 12) are import points but do not flow with the overall narrative.

R2: Following the suggestion of the referee, in order to sustain the coherency and flow in the discussion section, the first paragraphs containing information on effect of specimen size and critical cooling rate is inserted right after the description of the guiding rules for BMGs with acceptable or high GFA.

The paragraph about the embrittlement annealing is placed before the description of the origin of severe plasticity in Fe-based BMGs, where a consistent comparison of nanocrystal formation during casting and sub- T_g annealing process is now established.

Q3: Is there any evidence of spatial variations in nanocrystal concentration? This is suggested to be the origin of softer regions but no evidence of this spatial variation is provided.

R3: Following the suggestion of the referee, we probed various regions in the sample where different distribution of the nanocrystal clusters after autocorrelation analysis of FFT patterns are observed. The fraction of the nanocrystal clusters taken from an average thickness of 8 nm (bright zones in LM STEM (f)) is approximately 20 vol.%. Darker zones in LM STEM having lamellae thickness >20 nm (g) have nanocrystal cluster concentration as low as 10 vol.%. The clusters are more homogenously dispersed in the thinner regions than the thicker ones.

Figure – The spatial distribution of nanocrystal clusters retrieved by autocorrelation analysis of HRTEM images from lamellae thicknesses of (f) 8 nm and (g) >20 nm, and volumetric nanocrystal cluster fraction for each region in dimensions of 21 nm X 21 nm. The average volumetric fraction of the regions in (f) and (g) are $19.6 \pm 1.1 \mu\text{m}$ and $13.9 \pm 3.1 \mu\text{m}$, respectively.

We have placed a relevant text to the HRTEM Investigation of Nanocrystals section. The set of HRTEM images is embedded to Figure 5 (5f and 5g).

Q4: Please proofread for minor errors. For instance, on pg. 10 “There are numerous examples of BMGs of second nano- of 20%...”

R4: We apologize for this typo. The corrected is as follows: “There are numerous examples of BMGs with second phase of 20% or above in volume fraction formed with similar crystallization kinetics...”

Similarly, we checked the entire document for the final correction of other possible minor errors.

Q5: If possible, using a consistent unit for STZ size would be helpful. The main body uses nm while the theoretical calculation methods uses nm^3

R5: We would like to thank the referee for this recommendation. Because the mechanical behavior of BMGs is intrinsically depending on the actual volumes, based on the Johnson Samwer CSM model, STZ volume and the corresponding average number of atoms are calculated. Hence, the estimate STZ zone size of 2.02 nm is found for a cubic deformation region. We have now added the STZ zone size to the theoretical calculation section.